# Shedding Light on Heavy Metal Contamination: Fluorescein-Based Chemosensor for Selective Detection of Hg^2+^ in Water

**DOI:** 10.3390/ijms25063186

**Published:** 2024-03-10

**Authors:** Maksim N. Zavalishin, Alexey N. Kiselev, Alexandra K. Isagulieva, Anna V. Shibaeva, Vladimir A. Kuzmin, Vladimir N. Morozov, Eugene A. Zevakin, Ulyana A. Petrova, Alina A. Knyazeva, Alexey V. Eroshin, Yuriy A. Zhabanov, George A. Gamov

**Affiliations:** 1Faculty of Inorganic Chemistry and Technology, Ivanovo State University of Chemistry and Technology, 153000 Ivanovo, Russia; petrowa.ulya2015@yandex.ru (U.A.P.); knyazeva_alina1@mail.ru (A.A.K.); alexey.yeroshin@gmail.com (A.V.E.); zhabanov@gmail.com (Y.A.Z.); ggamov@isuct.ru (G.A.G.); 2G.A. Krestov Institute of Solution Chemistry, Russian Academy of Sciences, 153045 Ivanovo, Russia; scatol@yandex.ru; 3Burnazyan Federal Medical Biophysical Center, Federal Medical Biological Agency of the Russian Federtion, 123182 Moscow, Russia; kia2303@ya.ru; 4Institute of Gene Biology, Russian Academy of Sciences, 119991 Moscow, Russia; 5Emanuel Institute of Biochemical Physics, Russian Academy of Sciences, 119334 Moscow, Russia; anna-shiba@mail.ru (A.V.S.); vladimirkuzmin7@gmail.com (V.A.K.); morozov.v.n@mail.ru (V.N.M.); 6National Research Nuclear University MEPhI, 115409 Moscow, Russia; 7Vernadsky Institute of Geochemistry and Analytical Chemistry, Russian Academy of Sciences, 119991 Moscow, Russia; ezevakin.zev@yandex.ru

**Keywords:** fluorescein, hydrazone, fluorescent chemosensor, live cell imaging, mercury ion

## Abstract

This article discusses the design and analysis of a new chemical chemosensor for detecting mercury(II) ions. The chemosensor is a hydrazone made from 4-methylthiazole-5-carbaldehyde and fluorescein hydrazide. The structure of the chemosensor was confirmed using various methods, including nuclear magnetic resonance spectroscopy, infrared spectroscopy with Fourier transformation, mass spectroscopy, and quantum chemical calculations. The sensor’s ability in the highly selective and sensitive discovery of Hg^2+^ ions in water was demonstrated. The detection limit for mercury(II) ions was determined to be 0.23 µM. The new chemosensor was also used to detect Hg^2+^ ions in real samples and living cells using fluorescence spectroscopy. Chemosensor **1** and its complex with Hg^2+^ demonstrate a significant tendency to enter and accumulate in cells even at very low concentrations.

## 1. Introduction

Heavy metals are significant pollutants due to their ubiquitous presence in almost all natural ecosystems and their resistance to degradation processes, unlike organic pollutants [1]. Metal ions accumulate in soil easily, but their removal is difficult and slow, leading to their intensive accumulation in the tissues and organs of humans, living organisms, and hydrobionts [2,3]. Mercury occupies an unique position among heavy metals as its compounds are among the most toxic and hazardous substances [4]. The areas with the highest levels of mercury pollution in Russia are in proximity to metallurgical plants located in the Kola Peninsula, Urals, and Norilsk, where concentrations are several times higher than background levels [5,6]. The World Health Organization (WHO) defines the maximum allowable mercury concentration in drinking water as 0.006 mg/L [7]. Regular consumption of seafood increases the risk of methylmercury poisoning, as various studies worldwide have discovered [8]. Once in the body, the bloodstream easily transports mercury ions, leading to serious damage to the liver, kidneys, and brain [8,9,10]. Mercury salts commonly exist as divalent cations in an aqueous solution. Therefore, monitoring the Hg^2+^ ions in surface waters is a significant environmental safety task for numerous countries. 

Conventional methods like atomic absorption spectrometry (AAS) [11], ion chromatography (IC) [12], and optical emission spectroscopy with inductively coupled plasma (ICP) [13] have frequently been used to detect Hg^2+^ ions in the environment. The implementation of these techniques for on-site or real-time analyses is challenging due to the equipment’s high cost, intricate pre-processing procedures, and time-consuming nature. Optical techniques such as absorption spectroscopy and spectrofluorimetry rely on changes in color or fluorescence intensity and are not hindered by these problems [14,15]. Fluorescent chemosensors for Hg^2+^ ions are mainly derived from carbon dots [16,17], metal–organic frameworks [18,19], porphyrin [20], BODIPY [21,22], naphthalimide [23], coumarin [24], rhodamine [25,26], and fluorescein [27,28,29]. The development of new chemosensors for mercury ions is an important goal for researchers worldwide due to adverse environmental conditions.

Fluorescein is extensively used as a signaling component in fluorescent chemosensors because of its high quantum yield, photostability, and ease of functionalization [30,31,32]. A hydrazo group is frequently used as a connection between the receptor and the signaling fragment in fluorescein-based chemosensors [33,34]. Fluorescein hydrazones can be obtained with high yields and without generating by-products, making them a valuable asset for optical chemosensors. Recently, numerous fluorescent chemosensors for ions such as Cu^2+^ [35,36,37], Ni^2+^ and Al^3+^ [38], Zn^2+^ [39], Cd^2+^ [40], and Hg^2+^ [27,28,29] were developed based on fluorescein. As for Hg^2+^ chemosensors, 7-hydroxy-4-methylcoumarin-8-carbaldehyde [27], (pyridin-2-ylmethoxy)-naphthalene-1-carbaldehyde [28], and thiophene-2-aldehyde [29] were used as receptors. Based on the results of predicting the sensing ability of organic compounds towards metal ions based on their chemical formulae, hydrazone derived from fluorescein and 4-methylthiazole-5-carbaldehyde was found to be a potential chemosensor for Hg^2+^ ions [41] (Appendix A). The aim of this study is to broaden the choice of available chemosensors based on fluorescein molecules.

In the present article, we describe the synthesis and characteristics of a new sensor derived from 4-methylthiazole-5-carbaldehyde and hydrazide fluorescein (chemosensor **1**) to recognize mercury(II) ions. The ability of the compound to sense cations (Na^+^, K^+^, Ag^+^, Ca^2+^, Mg^2+^, Ba^2+^, Co^2+^, Ni^2+^, Cu^2+^, Zn^2+^, Cd^2+^, Pb^2+^, Hg^2+^, Cr^3+^, Fe^3+^, Ce^3+^, Al^3+^, In^3+^) was evaluated using UV-Vis and fluorescence experimental methods in an H_2_O/DMSO mixture. To supplement the experimental data, density functional theory (DFT) calculations were performed to determine the structure of **1** in solution. We also detected a strong enough fluorescence signal from the sensor with mercury ions in living cells, so it makes possible further work on the improvement of this sensor for application on living models. 

## 2. Results and Discussion

### 2.1. Selection of the Optimal H_2_O/DMSO Ratio for the Determination of Hg^2+^ Ions

Chemosensor **1** is insoluble in water, but soluble in many organic solvents (acetonitrile, alcohols, DMSO, THF, and DMF). Of these, DMSO is the most suitable for the determination of analytes because it has low toxicity to living organisms and is a high-boiling solvent, which makes it easier to prepare solutions of known concentrations. For these reasons, to select the optimal conditions for the determination of Hg^2+^ ions, we studied the fluorescence of the mixture **1**-Hg^2+^ in the H_2_O-DMSO system. Up to a 30 vol.% of water, chemosensor **1** exhibits no reaction with Hg^2+^ ions, likely due to the strong solvation of the cations by DMSO molecules [42]. Additionally, the opening reaction of the spirolactam ring of fluorescein derivatives occurs most efficiently in aqueous solutions [43]. Optimal fluorimetric conditions for the determination of mercury ions are observed in H_2_O-DMSO (8:2 *v*/*v*) (Figure 1). A sharp decrease in fluorescence intensity at 90 vol.% H_2_O is caused by aggregation of chemosensor **1**.

### 2.2. Selectivity of Chemosensor ***1*** towards Hg^2+^ and Other Cations in H_2_O-DMSO (8:2 v/v)

UV-Vis and fluorescence studies were performed using a 50 μM solution of chemosensor **1** in H_2_O-DMSO (8:2 *v*/*v*) with 5 eq. of common metal ions (Na^+^, K^+^, Ag^+^, Ca^2+^, Mg^2+^, Ba^2+^, Co^2+^, Ni^2+^, Cu^2+^, Zn^2+^, Cd^2+^, Pb^2+^, Hg^2+^, Cr^3+^, Fe^3+^, Al^3+^, Ga^3+^, In^3+^, and Ce^3+^) (Figure 2). 

Chemosensor **1** has two maxima in the UV region (276 and 336 nm). There are no absorption bands in the visible spectral region because of the closed spirolactam cycle, which makes the solution of **1** colorless (Figure 2a,c) [44]. However, when Hg^2+^ ions are added, the solution changes from transparent to brown, and a broad absorption band with a maximum of 448 nm appears. Additionally, fluorescence intensity at 520 nm significantly increases with the addition of 5 equiv. of Hg^2+^. The quantum yield also increases to 3.9% after the probe reacts with Hg^2+^ ions only, which demonstrates the excellent selectivity of chemosensor **1** for Hg^2+^ over other metal ions (Figure 2d). This selectivity can be utilized for the fluorescent determination of Hg^2+^ ions in solution. Fe^3+^ ions also induce the opening of the spirolactam cycle. However, their absorption is observed in the UV region and no fluorescence enhancement is observed (Figure 2c,d). 

Chemosensor **1** has a substantial affinity towards mercury ions, which renders hydrazone a fitting alternative for detecting Hg^2+^. Accordingly, the fluorescence response of **1** (50 μM) to Hg^2+^ ions (1 equiv.) mixed with some other metal ions (1 equiv.) in H_2_O-DMSO (8:2 *v*/*v*) is shown in Figure 3a. As can be seen from Figure 3, fluorescence intensity does not change significantly when 1 equiv. of various metal ions apart from Fe^3+^ is added to the chemosensor **1** solution. However, Fe^3+^ ions only interfere with the quantitative recognition of Hg^2+^ ions. Probably, Hg^2+^ and Fe^3+^ ions are competitors for opening the spirolactam ring of chemosensor **1.** Therefore, to determine Hg^2+^ in the presence of Fe^3+^ ions, it is necessary to mask the iron(III) ions. Fluorides are a suitable masking agent as they form a stable colorless complex [FeF_5_]^2−^. Chemosensor **1** exhibits high selectivity for Hg^2+^ ions, which is not hindered by most of the other metal ions in the H_2_O-DMSO mixture. Anions have a negligible impact on the fluorimetric detection of Hg^2+^ ions in solution, except halides such as Cl^−^, Br^−^, and I^−^ (Figure 3b). According to the literature, Hg^2+^ ions can form stable complexes of the composition [HgHal_n_]^2−n^ with halides [45,46,47]. Such anions as Cl^−^, Br^−^, and I^−^ are competing with chemosensor **1** for Hg^2+^ ions in solution. However, qualitative determination of mercury(II) ions is possible, when Hg^2+^ ions and halides (Cl^−^, Br^−^, or I^−^) are equimolar in solution. 

Adding increasing amounts (up to 180 µM) of Hg^2+^ ions (Figure 4a) leads to an increase in the intensity of the emission band, with a maximum at 520 nm in the fluorescent spectra of chemosensor **1**. This increase in fluorescence is attributed to the formation of a coordination compound with a higher quantum yield in comparison to chemosensor **1**. We attribute such changes to opening the spirolactam ring of the chemosensor by complexation. The mechanism of spirolactam opening in xanthene dyes, leading to fluorescence turn-on, is well known [48,49]. The stability constant of the **1**-Hg^2+^ complex was computed from the spectrofluorimetric data (Figure 4) using KEV software, version 0.7 [50], while the stoichiometry of the **1**-Hg^2+^ complex was determined via mass spectrometry (Appendix A). Furthermore, the best description of the experimental fluorimetric data is observed for the 1:1 stoichiometric model (R^2^ = 0.958) (Appendix A). The stoichiometric model 1:2 is suboptimal, leading to the conclusion that the formation of a 1:2 complex is improbable (Appendix A). The conditional stability constant of **1**-Hg^2+^ lg β = 3.54 ± 0.24 was determined using KEV software, version 0.7 [50] (Appendix A). A possible reaction scheme between chemosensor **1** and the Hg^2+^ ion is shown in Appendix A. The detection limit of chemosensor **1** for Hg^2+^ was also calculated from the titration to be 0.23 μM (Appendix A), which was lower than the majority of the reported LOD values for Hg^2+^ chemosensors (Appendix A). 

Figure 5a shows that the absorption properties of chemosensor **1** remain stable for 8 days, proving its practicality for Hg^2+^ recognition purposes. Upon the addition of Hg^2+^ to the chemosensor **1** solution, the fluorescence intensity quickly approached its peak after 100 min and remained stable thereafter (indicated by red dots). This demonstrates the chemosensor’s rapid detection capability for mercury(II) ions.

### 2.3. Practical Application

We tested the practical usefulness of chemosensor **1** by examining water samples from local rivers for Hg^2+^ ions. The water was collected from the Uvod (57°00′09.0″ N 40°59′46.7″ E) and Uhtokhma rivers (56°89′62.7″ N 40°63′29.6″ E) in the Ivanovo region, Russia. The samples were filtered to remove solids and then treated with a standard solution containing different concentrations of Hg^2+^ ions. The experiment was conducted three times to ensure accuracy. Appendix A displays an instance of the spectral measurements of chemosensor **1** when Hg^2+^ ions were present in the sample. The results, shown in Table 1, indicate that chemosensor **1** effectively identifies Hg^2+^ ions in real samples. 

It is also possible to qualitatively determine mercury(II) ions using a UV lamp. Yellow-green fluorescence is only observed when Hg^2+^ ions are in solution (Figure 2b).

### 2.4. Molecular Structure and Electronic Spectra of Chemosensor ***1***

Electronic absorption spectra play an important role in the design of chemosensors. Quantum chemical (QC) calculations are a powerful tool for simulating and then understanding the nature of electronic absorption spectra. It is also important to take into account that compounds can exist as several forms (conformers), which affect reactivity and spectral characteristics.

Quantum chemical calculations showed the coexistence of two conformers (**1a** and **1b**) of **1** at room temperature (RT, T = 298 K), differing by an orientation of methylthiazole fragment (Figure 6). Both conformers possess a C_s_ symmetry point group, i.e., there is a plane of symmetry including the atoms of thiazole and isoindoline rings. The contributions of **1a** and **1b** are directly proportional to $e^{\frac{-∆G_{i}}{RT}}$ and were evaluated as:$$x_{i}=\frac{e^{\frac{-∆G_{i}}{RT}}}{\sum e^{\frac{-∆G_{i}}{RT}}}$$
where $∆G_{i}$ is the relative Gibbs energy of the conformer. So, **1a** dominates in equilibrium at RT; its amount is 96 mol. %. The transition from **1a** to **1b** can be achieved by rotating the thiazole ring around the hydrazone bridge with a barrier of ~7.7 kcal/mol (numbers of atoms correspond to those in the scheme for synthesizing). The potential energy surface profile of such a rotation is depicted in Figure 6; enlarged images of molecular models of **1a**, **1b**, and transition state (**TS**), along with their relative energies and contributions, can be found in ESI (Appendix A).

Since the simulated electronic absorption spectra of **1a** and **1b** are close to each other (maximal difference in band position is 4 nm, Figure 7), we describe below only the spectrum of **1a**, which is favorable according to QC calculations. The highest occupied molecular orbital (HOMO, Figure 8) with a” symmetry is localized on the bonding π-orbitals of the thiazole and isoindoline rings and the C-N-bridge between them—however, without orbitals of the sulfur atom. The most intensive peak in the spectra is at 319 nm and corresponds to electronic transitions from HOMO to the lowest unoccupied molecular orbital (LUMO). The latter includes π-orbitals centered on the same rings as HOMO but with sulfur atomic orbitals. So, the band at 319 nm is caused by π → π* transitions in these fragments. The calculated composition of the lowest excited states and corresponding oscillator strengths are listed in Appendix A. Shapes of orbitals involved in electronic transitions are represented in Appendix A.

### 2.5. Cytotoxicity of ***1*** and ***1***:Hg^2+^ Complex

The detection of mercury and its ions in living organisms may be very useful for various medical and research tasks due to their danger to the metabolism. Potential chemosensors must be both neutral to biological systems and highly sensitive, to detect very small amounts of target ions. Thus, to investigate possible harmful effects, **1** and its complex with Hg^2+^ were tested using the MTT assay (Figure 9, Table 2). Compound **1** is found toxic enough to both tumor (HCT116) and non-tumor (HEK293T) cell lines when incubated for 3 days at a concentration of 20 μM or more. Half-maximal inhibitory concentrations (IC_50_) in these cases are about 21 and 20 μM, respectively. In turn, the cytotoxicity of the **1**:Hg^2+^ complex is a little higher: the corresponding IC_50_ values are about 15 and 16 μM. These results partly limit the application of the **1** compound as a chemosensor for living systems but pose the task of further modifying it in order to reduce such negative effects.

### 2.6. Detection of ***1*** and ***1***:Hg^2+^ Complex in Living Cells

Molecules of the **1**:Hg^2+^ complex are successfully accumulated by HCT116 cells in a quantity that is sufficient for fine fluorescence detection using flow cytometry. Although concentrations of 25 and 50 μM are lethal for 50% of the cell population after 3 days of exposition, a short incubation (1.5 or 24 h) allows us to visualize cellular uptake and the accumulation of the complex with a strong signal (Figure 10A). In non-toxic concentrations up to 2 μM, the signal is much weaker even after 24 h; however, the complex is still detectable (see green and pink lines in Figure 10A).

The difficulty is to estimate the intracellular uptake and long accumulation of **1** alone (Figure 10B, pink and green lines) because of its weak fluorescence and solubility properties. At the same time, we have shown that 4 h preincubation with 25 μM of **1**, followed by the addition of 50 μM Hg^2+^, almost does not increase the cellular accumulation of the newly formed complex after 24 h of total incubation (see orange line in Figure 10B). This fact may also be explained by a poor reaction of **1** and Hg^2+^ in the cultural medium.

We also investigated the intracellular fluorescence of **1** and the **1**:Hg^2+^ complex by confocal microscopy. As shown in Figure 11, the **1**:Hg^2+^ complex is well visualized in HCT116 cells both at chemosensor concentrations of 10 and 25 μM. The fluorescence signal of compound **1** at a concentration of 10 μM is not registered in cells. At a concentration of 25 μM, compound **1** shows extremely weak, almost invisible fluorescence. Accumulation of the **1**:Hg^2+^ complex occurred mainly in the cytoplasm (there is also the possibility of accumulation in the mitochondria, lysosomes, and endoplasmic reticulum) and, to a much lesser extent, in the cell nucleus. Summarizing the data of in vitro experiments, chemosensor **1** and its complex with Hg^2+^ have the fundamental ability to permeate and accumulate in cells even in a nanomolar concentration; however, the cells accumulate compound **1** alone at a much lower level than its complex with mercury. Thus, despite relatively high cytotoxic effects for tumor and non-tumor cells, the tested compound is of great interest for future investigation as a fluorescence probe of mercury ions.

## 3. Materials and Methods

### 3.1. Chemicals

4-methylthiazole-5-carbaldehyde and fluorescein hydrazide (BLD Pharm, Telangana, India) were utilized in their pure form. The manufacturers claimed the purity of the reagents to be >98 wt.%. Nitrate salts of Na^+^, K^+^, Ag^+^, Ca^2+^, Mg^2+^, Ba^2+^, Co^2+^, Ni^2+^, Cu^2+^, Zn^2+^, Cd^2+^, Hg^2+^, Pb^2+^, Cr^3+^, Fe^3+^, Al^3+^, Ga^3+^, In^3+^, and Ce^3+^ procured from Reakhim (Staraya Kupavna, Russia), were used as received. Sodium salts of F^−^, Cl^−^, Br^−^, I^−^, CN^−^, NCS^−^, NO_3_^−^, ClO_3_^−^, SO_3_^2−^, H_2_PO_4_^−^, HSO_4_^−^, AcO^−^, and ClO_4_^−^ procured from Reakhim (Staraya Kupavna, Russia), were also utilized without purification. Solutions of cation and anion salts were prepared using deionized water (conductivity of κ = 3.6 μS/cm and pH = 6.6). Commercially available dimethyl sulfoxide and ethanol (EKOS-1, Staraya Kupavna, Russia, purity > 99.9% and 95.58%) were also incorporated.

### 3.2. Synthesis of Chemosensor ***1***

Solutions of 4-methylthiazole-5-carbaldehyde (0.1281 g, 1.0 mmol) and fluorescein hydrazide (0.3466 g, 1.0 mmol) in ethanol were mixed in a flask. The resultant mixture was refluxed and stirred for 6 h. Upon cooling to room temperature, a finely dispersed pale grey precipitate formed. The obtained crystalline product was filtered, washed with ice-cold ethanol and acetone, and dried at 40 °C to a constant weight. The yield was 0.3097 g (68%). The synthesis pathway for chemosensor 1 is depicted in Figure 12.

(E)-3′,6′-dihydroxy-2-((4-methylthiazol-5-yl)methyleneamino)spiro[isoindoline-1,9′-xanthen]-3-one: ^1^H NMR, δ, ppm (DMSO-d6): 9.95s (2H, OH), 9.08s (1H, H_2_), 8.94s (1H, H_6_), 7.92d (^3^J = 7.4 Hz, 1H, H_8_), 7.66t (^3^J = 7.0 Hz, 1H, H_10_), 7.61t (^3^J = 7.0 Hz, 1H, H_9_), 7.14d (^3^J = 7.4 Hz, 1H, H_11_), 6.65d (^4^J = 1.1 Hz, 2H, H_18_), 6.48d (^4^J = 1.1 Hz, 4H, H_15,16_), 2.29s (3H, H_4′_). ^13^C NMR, δ, ppm (DMSO-d6: 164.0 (C=O), 159.2 (C_17_), 155.7 (C_2_), 154.7 (C_19_), 152.7 (C_4_), 150.8 (C_12_), 141.2 (C_6_), 134.6 (C_7_), 129.7 (C_10_), 129.3 (C_15_), 128.6 (C_5_), 128.6 (C_8_), 124.4 (C_11_), 123.7 (C_9_), 112.9 (C_16_), 110.1 (C_14_), 102.8 (C_18_), 65.7 (C_13_), 15.5 (C_4′_). *m*/*z* = 456.55 [1+H^+^], theoretical *m*/*z* = 456.49 [1+H^+^]. IR, (KBr) cm^−1^: 3422s, 2921m, 2851s, 1701s, 1669s, 1614s, 1505s, 1446s, 1390m, 1312s, 1269s, 1172s, 1113s, 994m, 849m, 758m, 692m, 625w, 475w.

IR, MS, ^1^H, and ^13^C NMR spectra are given in the Appendix A.

### 3.3. Spectral Measurements

The UV-Vis spectra were measured using a double-beamed Shimadzu UV1800 spectrophotometer (Shimadzu, Somerset, NJ, USA). The wavelength range was 260–700 nm and the absorbance range was 0–1 in H_2_O-DMSO (8:2 *v*/*v*). H_2_O-DMSO (8:2 *v*/*v*) was used as a blank solution. The temperature was kept constant at 298.2 ± 0.1 K with an external thermostat.

Fluorescence spectra were recorded using the RF6000 setup (Shimadzu, Somerset, NJ, USA) with an excitation wavelength of λ_ex_ = 446 nm and an emission wavelength range of 470–750 nm. The excitation and emission slit widths were set to 5 nm. The temperature was maintained at 298.2 ± 0.1 K using an external thermostat. 

Three-dimensional fluorescence spectra were obtained by recording the spectra within the range of λ_ex_ = 300–600 nm and λ_em_ = 320–800 nm (Figure 13). The optimal excitation wavelength for detecting Hg^2+^ with chemosensor **1**, which results in strong emission intensity, was found to be λ_ex_ = 446 nm.

The NMR experiments for chemosensor **1** were conducted on a Bruker Avance III 500 NMR spectrometer (Bruker, Billerica, MA, USA) with 500.17 MHz and 125.77 MHz frequencies for ^1^H and ^13^C, respectively, in DMSO-*d6*. Temperature control was maintained using a Bruker variable temperature unit (BVT-2000), and the experiments were carried out at 298 K without sample spinning. The accuracy of the chemical shift measurement was determined to be ±0.01 ppm for ^1^H NMR spectra and ±0.1 ppm for ^13^C NMR, according to the external standard, HMDSO. 

The MS (MALDI TOF) spectra for chemosensor **1** and **1**+Hg^2+^ were obtained using the Shimadzu Biotech Axima Confidence system, which was produced by Shimadzu in the NJ, United States. The samples were dissolved in an EtOH-H_2_O mixture and applied to the plate. The samples were then allowed to air-dry before the experiment.

The Avatar 360 FTIR spectrometer, manufactured by Thermo Nicolet in the MA, United States, was used to record the IR spectra for chemosensor **1**. The sample was dispersed in KBr, and a range of 400–4000 cm^−1^ was scanned. 

### 3.4. Computational Details

The Gaussian 09 [51] program was used for geometry optimization, followed by harmonic frequencies calculation. The calculations were carried out using density functional theory (B97D functional [52]) with a triple-zeta def2-TZVP basis set [53] taken from the EMSL BSE library [54,55,56]. Basis sets for S, O, and N atoms were also augmented by diffuse functions (def2-TZVPD [53,57]). Electronic absorption spectra were calculated using the time-dependent density functional theory approach. The number of excited states was 30. The polarizable continuum model (PCM, the solvent is water) was applied to take into account solvation effects in electron absorption spectra calculations. Cartesian coordinates of quantum chemical structures are presented in Appendix A.

### 3.5. Cells

Embryonic human kidney cells (HEK293T) and human colon adenocarcinoma (HCT116) were obtained from the American Type Culture Collection (ATCC, Manassas, VA, USA). Cells were cultivated at 37 °C in a 5% CO_2_ atmosphere. For this, DMEM (PanEco, Moscow, Russia) supplemented with 10% FBS (Cytiva, Marlborough, MA, USA), 2 mM L-glutamine (PanEco, Moscow, Russia), 100 U/mL penicillin (PanEco, Russia), and 100 μg mL^−1^ streptomycin (PanEco, Moscow, Russia) was used. 

### 3.6. MTT Assay

The cytotoxicity of compounds was measured by using the MTT assay (MTT is an abbreviation for the dye compound 3-(4,5-Dimethylthiazol-2-yl)-2,5-diphenyltetrazolium bromide). Doxorubicin (Veropharm, Moscow, Russia) was taken as a control. Cells were seeded in 96-well plates in a density of 5000 cells per well. At 24 h after attachment, the compounds were added at a different concentration for the next 72 h. After incubation, 5 mg/mL of MTT reagent (Dia-M, Moscow, Russia) was added to each well; then, the plates were incubated for another 3 h under the same conditions. The precipitate of formazan crystals was dissolved in 100 μL DMSO. Colorimetric measurement was performed at λ = 570 nm using a CLARIOstar Plus microplate reader (BMG LABTECH, Ortenberg, Germany). The experiment was repeated at least 3 times.

### 3.7. Flow Cytometry

The accumulation of **1** and its complex with Hg^2+^ in cells was investigated using flow cytometry. HCT116 cells were seeded on 35 mm Petri dishes in a density of 300,000 cells per dish. At 24 h after attachment, compounds were added to the culture; then, the cells were incubated for a different time. Before measuring, the cells were rinsed with PBS, detached by Versene solution (PanEco, Moscow, Russia), and centrifuged at 500 rcf for 5 min. Cell pellets were resuspended and rinsed twice in fresh PBS and analyzed using a CytoFLEX flow cytometer (Beckman Coulter, Brea, CA, USA) in FITC channel (λ_ex_ = 488 nm, λ_em_ = 525/40 nm).

### 3.8. Confocal Microscopy

HCT116 cells were seeded in a density of 100,000 cells per well on 35 mm confocal dishes (Eppendorf, Hamburg, Germany) and were incubated at 37 °C in 5% CO_2_ for 24 h. Then, the compound under assay was added and cells were incubated for another 4 h under the same conditions. The fluorescence in PBS was excited at λ_ex_ = 458 nm and detected in the wavelength range of 490–560 nm. The signal was normalized by the control dish without the compound. Samples were recorded on the laser scanning confocal microscope Leica TCS SPE 5 with LAS AF software, version 2.6 (Leica Microsystems GmbH, Wetzlar, Germany). 

## 4. Conclusions

A novel fluorescent chemosensor designed for detecting mercury(II) ions, hydrazone derived from 4-methylthiazole-5-carbaldehyde and fluorescein hydrazide was synthesized and characterized using various spectral methods and quantum chemical calculations. According to the quantum chemical calculations results, chemosensor **1** exists in the form of two conformers differing by an orientation of the methylthiazole fragment; however, the electronic absorption spectra of these conformers are almost the same. Chemosensor **1** exhibited increased emission intensity in its H_2_O-DMSO (8:2 *v*/*v*) solution in the presence of Hg^2+^ ions, while the addition of other ions such as Na^+^, K^+^, Ag^+^, Ca^2+^, Mg^2+^, Ba^2+^, Co^2+^, Ni^2+^, Cu^2+^, Zn^2+^, Cd^2+^, Pb^2+^, Cr^3+^, Fe^3+^, Al^3+^, Ga^3+^, In^3+^, and Ce^3+^ did not affect its intrinsic fluorescence. The interference of different cations was also investigated, and a qualitative analysis of Hg^2+^ ions was shown to be possible in the presence of most cations in the sample. The detection limit (LOD = 0.23 µM) for chemosensor **1** was determined and compared with the literature data. Chemosensor **1** was demonstrated to be applicable for monitoring the mercury(II) concentration in real samples with a high recovery rate. Despite of a relatively high cytotoxic effect for tumor and non-tumor cells, chemosensor **1** is of great interest for future investigation as a fluorescence detector of mercury ions. Future work will focus on expanding the range of cations that can be analyzed using fluorescein hydrazones.

## Figures and Tables

**Figure 1 ijms-25-03186-f001:**
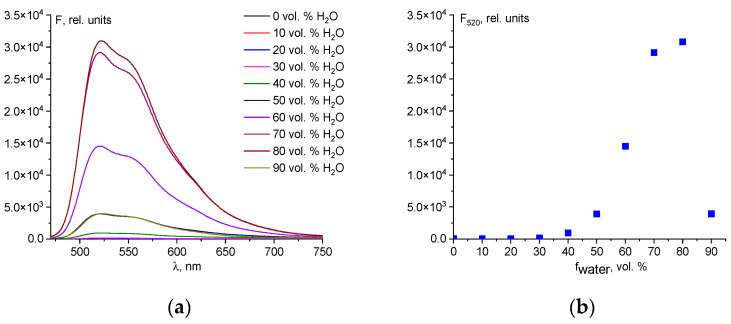
Fluorescence spectra of the mixture of chemosensor **1** (50 µM) and Hg^2+^ (250 µM) in H_2_O/DMSO mixtures with different water fractions (**a**); The dependence of the fluorescence intensity at the emission maximum of 520 nm for a mixture of chemosensor **1** (50 µM) and Hg^2+^ (250 µM) in H_2_O/DMSO mixtures (**b**).

**Figure 2 ijms-25-03186-f002:**
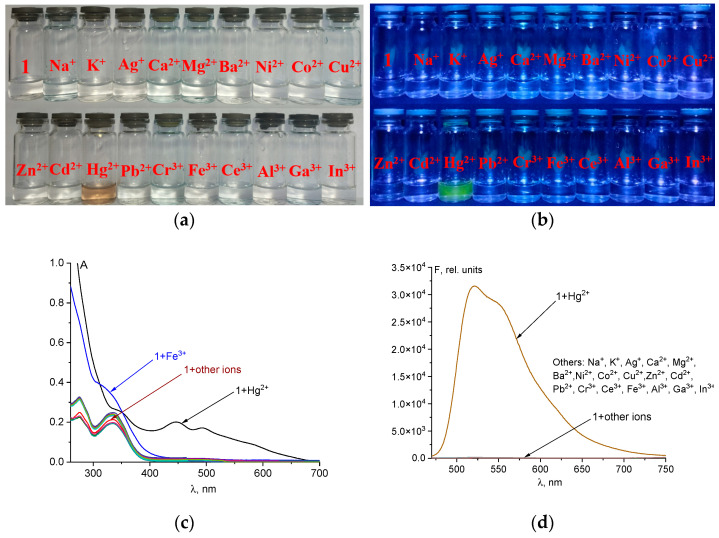
Solution color (**a**), naked-eye visible luminescence at λ_ex_ = 365 nm (**b**), UV-Vis spectra (**c**), and fluorescence spectra (**d**) of chemosensor **1** (50 μM) and its mixtures with different cations (250 μM) in H_2_O-DMSO (8:2 *v*/*v*).

**Figure 3 ijms-25-03186-f003:**
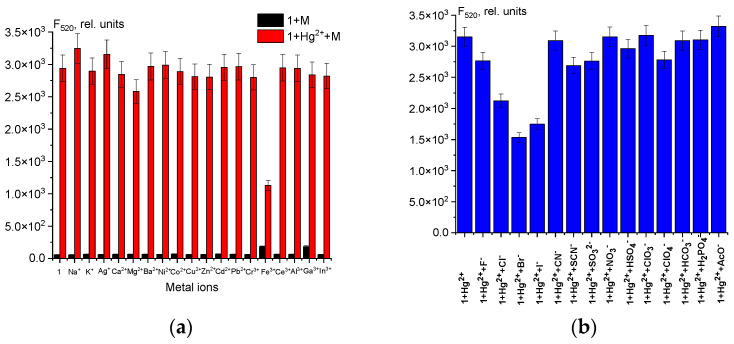
Fluorescence intensity at λ_em_ = 520 nm of **1** + cations and **1**-Hg^2+^ + cations mixtures (**a**); emission spectra of **1**-Hg^2+^ + anions mixtures (**b**). The solvent is H_2_O-DMSO (8:2 *v*/*v*).

**Figure 4 ijms-25-03186-f004:**
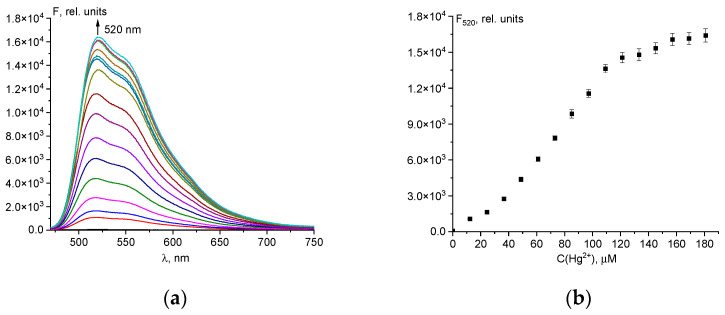
Spectrofluorimetric titration of chemosensor **1** (50 µM) by 0.01103 M Hg^2+^ ions solution (**a**); dependence of fluorescence intensity at λ_em_ = 520 nm on the total concentration of mercury(II) ions (**b**). The solvent is H_2_O-DMSO (8:2 *v*/*v*). The error bars indicate the random inaccuracy for the three experiments.

**Figure 5 ijms-25-03186-f005:**
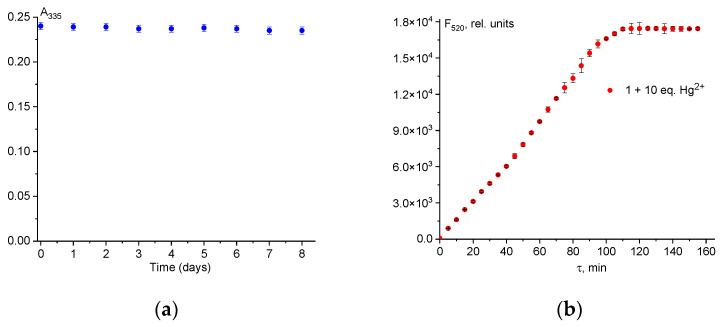
Absorption change at λ_em_ = 335 nm of chemosensor **1** (50 µM) at 335 nm during 8 days (**a**); fluorescence intensity change at λ_em_ = 520 nm of mixture **1** (50 µM) + (500 µM) Hg^2+^ during 160 min (**b**). Solvent is H_2_O-DMSO (8:2 *v*/*v*). The error bars indicate the random inaccuracy for the three experiments.

**Figure 6 ijms-25-03186-f006:**
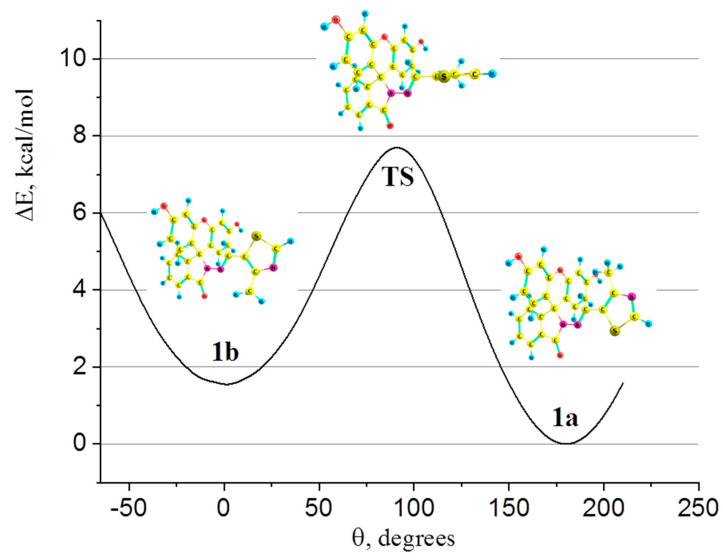
Potential energy surface profile of the thiazole ring rotation; θ is N-C5-C6-C4 torsion angle.

**Figure 7 ijms-25-03186-f007:**
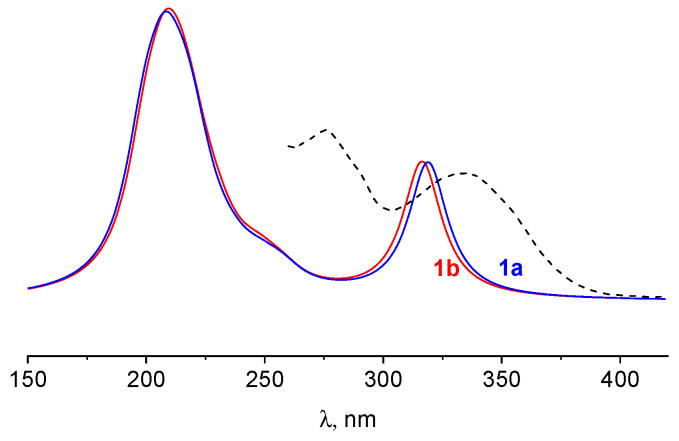
Calculated (solid lines) and experimental (dashed line) electronic absorption spectra of **1**.

**Figure 8 ijms-25-03186-f008:**
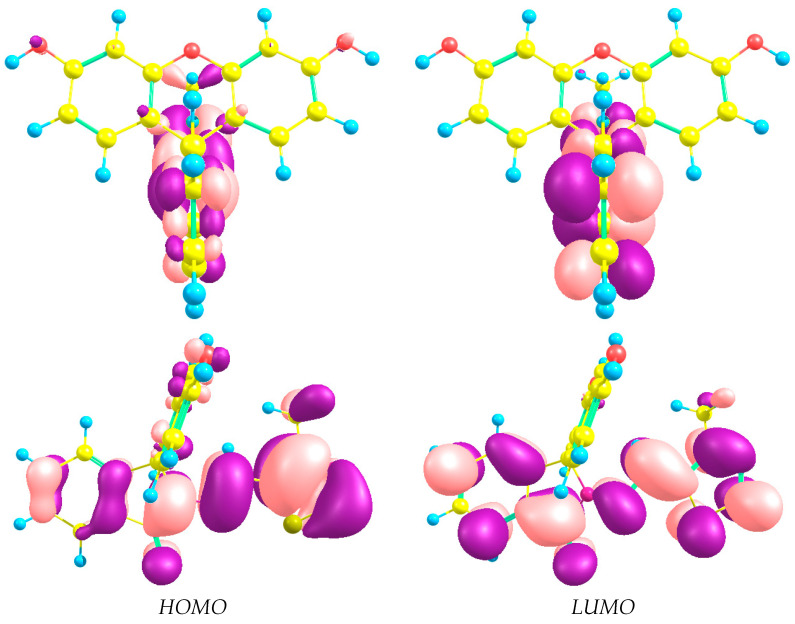
Shapes of frontier orbitals of **1**.

**Figure 9 ijms-25-03186-f009:**
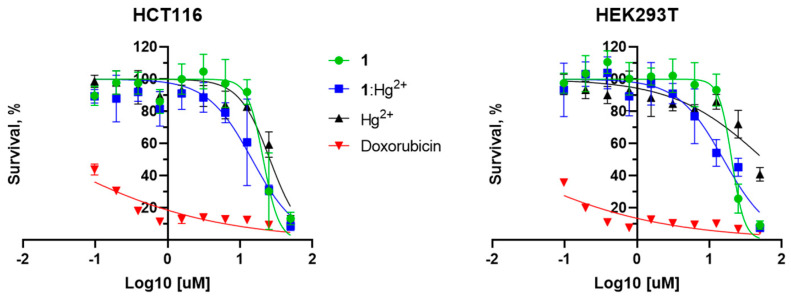
Measurements of cytotoxicity of **1** compound (green) and Hg^2+^ salt (black), as well as their complex (blue), to human colon carcinoma cells (HCT116) and non-tumor human embryonal kidney cells (HEK293T) after 72 h of incubation. Doxorubicin (red) was taken as a standard for the comparison.

**Figure 10 ijms-25-03186-f010:**
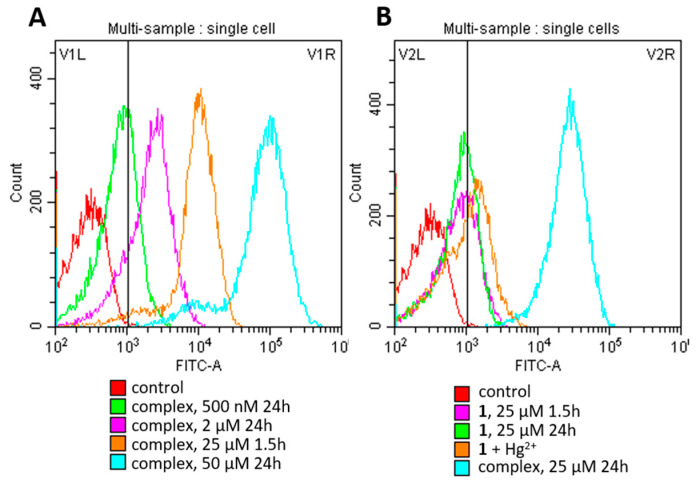
Accumulation of **1**:Hg^2+^ complex in HCT116 cells at various concentrations and incubation times (**A**), as well as of compound **1** alone followed by the addition of Hg^2+^ ions (**B**).

**Figure 11 ijms-25-03186-f011:**
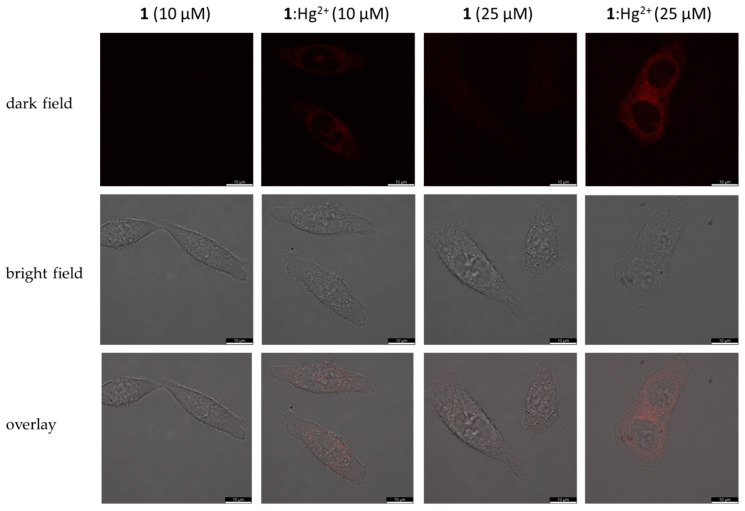
Confocal microscopy of HCT116 cells after 4 h of incubation with compound **1** (50 μM) or **1**:Hg^2+^ complex (25 and 50 μM): dark field (**left**), BF—bright field (**middle**), and overlay (**right**).

**Figure 12 ijms-25-03186-f012:**
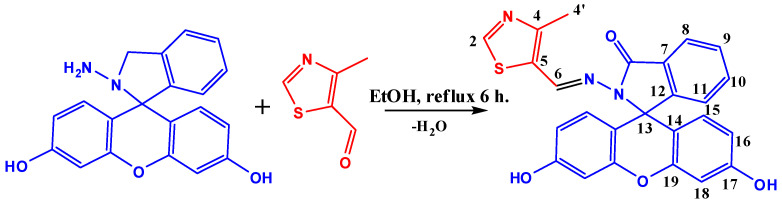
Scheme for synthesizing chemosensor **1**.

**Figure 13 ijms-25-03186-f013:**
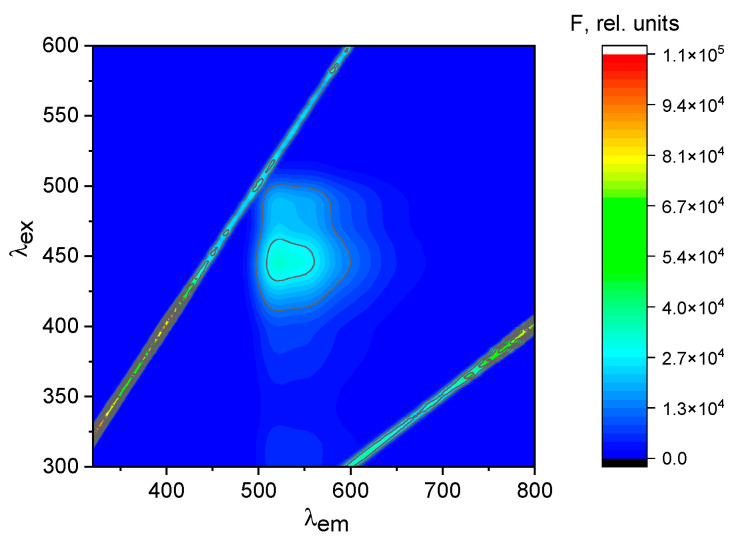
Contour map λ_ex_–λ_em_ chemosensor **1**-Hg^2+^ mixture (50:250 µM).

**Table 1 ijms-25-03186-t001:** Determination of Hg^2+^ ions in various water samples.

Sample	Added, µM	Found, µM	Recovery, %
Uvod river	30.00	28.74 ± 2.12	95.80
	40.00	37.55 ± 2.91	93.88
	60.00	54.93 ± 3.63	91.55
Uhtokhma river	30.00	29.01 ± 1.84	96.70
	40.00	37.18 ± 2.54	92.95
	60.00	57.43 ± 2.32	95.72

**Table 2 ijms-25-03186-t002:** Half-maximal inhibitory concentrations (IC_50_) of **1**, Hg^2+^ salt and **1**:Hg^2+^ complex.

	1	1:Hg^2+^	Hg^2+^
HCT116	20.96	14.68	25.91
HEK293T	20.38	15.74	57.17

## Data Availability

All the data used are given either in the text of the present paper or the Appendix A.

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
