# Peer review of "Shedding Light on Heavy Metal Contamination: Fluorescein-Based Chemosensor for Selective Detection of Hg2+ in Water"

_ijms, 2024, doi:10.3390/ijms25063186_

Round 1

Reviewer 1 Report

Comments and Suggestions for Authors

In this work, a fluorescein-based chemosensor was developed for selective detection of Hg2+ in water. The structure of the chemosensor was confirmed using various methods including characterization spectroscopy and quantum chemical calculations. However, the quality of this manuscript should be significantly improved, there were many points need to be improved. I do not recommend its publication in this journal.

Points for further revision:

1.       To facilitate the reader's understanding,it is better to add the schematic diagram of bonding between chemosensor 1 and Hg2+.

2.       The proposed chemosensor exhibited unusual response Fe3+ ions, this should be discussed clearly, and some methods should be provided to avoid the interference to Hg2+ sensing.

3.       The error bars were lacked in Figure 4 and Figure 5,please add them and explain in figure captions.

4.       The LOD of this proposed chemosensor was 0.23 μM, not lower than most reported methods, this should be discussed reasonably.

5.       The resolution of figures should be improved, and the figures should be adjusted professionally, especially figures in Supporting Information.

6.       The whole manuscript should be checked carefully and correct some mistakes, as well as Supporting Information.

7.       More related papers should be referenced for strengthen the research background.

Comments on the Quality of English Language

Extensive editing of English language required

Reviewer 2 Report

Comments and Suggestions for Authors

Maksim et al prepared a manuscript entitled “Shedding Light on Heavy Metal Contamination: A Fluorescein- 2 Based Chemosensor for Selective Detection of Hg2+ in Water” for submission under International Journal of Molecular Sciences. In this study, the authors prepared chemosensor and the sensor's ability was checked by UV-Vis and fluorescence experimental methods. The practical application was made using real samples. After careful evaluation of the manuscript, I have decided to  recommend this work to be published in the well reputed “International Journal of Molecular Sciences” journal for the publication. The following are the major comments to be addressed before being considered for publication.

1.   Line 76: It is mentioned that “In addition, the hydrazone produced was found to be suitable for the recognition of Hg2+ in living cells”, but the suitable experimental results are not provided. Clarify it.

2.   For a valid fluorescence sensor, one must study the mechanistic pathway. This study did not show any experimental data related to the mechanism involved. Describe it in detail with valid results.

3.   Describe how to correlate the mercury sensing with the flow cytometry experimental studies?

4.   What is the purpose of considering HEK293T and HCT116 cell lines when analysing mercury detection?

5.   The data inferred from confocal experiment did not provide valid information on live cell studies. The experiment should be corrected accordingly.

Round 2

Reviewer 1 Report

Comments and Suggestions for Authors

From my view of point, the manuscript can be considered for publication at the status because it was well revised.

Comments on the Quality of English Language

From my view of point, the manuscript can be considered for publication at the status because it was well revised.